# The Impact of Social Capital on Rural Residents’ Medical Service Utilization in China—An Empirical Study Based on CFPS Data

**DOI:** 10.3390/ijerph192315989

**Published:** 2022-11-30

**Authors:** Hongyun Zhou, Jiqing Hong, Su Yang, Yuxuan Huang

**Affiliations:** 1School of Public Administration, Zhongnan University of Economics and Law, Wuhan 430073, China; 2School of Environment, Education and Development, The University of Manchester, Manchester M13 9PL, UK

**Keywords:** rural residents, social capital, medical service utilization, rural revitalization, medical institution

## Abstract

Based on CFPS data, the article analyzes the impact of social capital on the utilization of medical services by rural residents in China using binary logit and multinomial logit models. The social capital includes two dimensions: cognitive social capital and structural social capital, and the indicators of medical service utilization are “whether to seek medical treatment when sick” and “choice of medical institution”. This paper concludes that: (1) among the cognitive social capital, special trust and religious beliefs have a significant positive influence on whether rural residents choose to seek medical treatment when they are sick; (2) among the structural social capital, social participation has a significant positive influence on choice of medical institution, and social network has a significant positive influence on the choice of township health center, specialty hospital, and general hospital. According to the empirical results, this paper proposes the following suggestions. The trust mechanism of rural society should be reconstructed, the positive role of religious beliefs should be given full play, the healthy development of social networks should be promoted, and the rural social organizations should be fostered and developed.

## 1. Introduction

Health is an important issue of concern to the world, and China also puts high emphasis on people’s health. Among the influencing factors of health, medical service utilization is one of the most important factors. In order to provide the people with equitable health protection, China has issued *“Healthy China 2030” Planning Outline,* taking measures to continuously improve people’s health. Equitable access and utilization of health care services is a central goal of the health care system [1]. Enhancing the utilization level of medical services in rural areas is an important measure to guarantee the health of rural residents.

However, there is still a problem of unbalanced development of medical services between urban and rural areas in China. It is necessary and urgent to enhance the utilization level of medical services in rural areas. *“Law of the People’s Republic of China on the Promotion of Rural Revitalization”* clearly proposes to “promote the inclination of medical and health resources to rural areas, enhance the level of basic public services in rural areas, and promote the equalization of basic public services between urban and rural areas”. Health care services should be provided according to the principle of horizontal equity, with priority given to those who need it most [2]. To further improve the utilization of medical services and meet the health needs of rural residents is not only an objective requirement of the “Health China” strategy but also an important way to consolidate and expand the achievements of poverty alleviation.

Social capital is an important factor affecting the utilization of medical services. Good social capital can broaden people’s access to medical information and improve their ability to obtain medical resources [3], thus helping them to make better use of medical services. Especially in rural China, which is mainly characterized by “acquaintance society”, social capital is a key factor influencing rural residents’ access to information about medical resources and medical services.

In the book “*From the Soil—The Foundations of Chinese Society*”, Mr. Fei Xiaotong proposed the concept of “acquaintance society” based on empirical facts, in which villagers are born and die, forming a familiar society without strangers [4]. Despite the impact of marketization and urbanization on the traditional rural “acquaintance society”, social capital is still an important factor affecting the utilization of medical services by rural residents. The reason are as follows: First, compared with urban areas, information channels in rural areas are relatively homogeneous and insufficient. Unlike urban residents who can collect medical information through multiple channels such as online platforms, advertisements, and bulletin boards, most rural residents mainly rely on “word-of-mouth” to obtain information. In addition, the overall education level of rural residents is not high, and their ability to identify, judge, and make decisions about medical services is relatively weak. Therefore, villagers trust “recommendations from acquaintances” more than urbanites.

The resources inherent in the social capital can prevent physical diseases [5,6]. However, a review of the existing literature shows that studies on the influencing factors of medical services utilization scarcely focus on social capital. Andersen proposed a model of medical service utilization, pointing out that the utilization of health care services by individuals is mainly influenced by three types of factors [7]. First is the propensity factor, which refers to certain characteristics that lead to illness, such as older age and unhealthy lifestyle. Second is the need factor, which refers to the physical characteristics that lead to medical behavior, such as poor health, chronic diseases, disability, etc. Third is the ability factor, which refers to the ability of individuals to access health care services, including income level, insurance participation, etc. [8]. Medical insurance is an important influencing factor of medical or health services. The research of the RAND Corporation showed that medical insurance had an impact on whether patients seek medical treatment [9,10]. From the perspective of the relationship between medical policies and medical service utilization, some scholars have proposed that different medical services have cross-price elasticity with each other [11,12]. Feldstein proposed that the demand for medical insurance and the price of medical services interact and spiral [13].

Family factors mainly include intergenerational support and family care. Brewster found that higher family support was significantly associated with lower readmission rate [14]. William showed that family care led to a different type of health care [15]. Yi found family care can improve the utilization level of prenatal care for itinerant pregnant women [16]. In addition, other factors such as marital status [17] and visible work income state [18] cause effects directly or indirectly.

Individual factors are also critical to medical service utilization. Chernichovsky proposed that income impacted the utilization of medical services [19]. The hukou system may hinder the inequality in the utilization of medical services [20]. Regarding personal education, there are different results from different perspectives. Eiko found that individuals with higher health literacy have better health outcomes and use emergency rooms and hospital services less frequently [21,22]. On the other hand, Dunlop [23] and Terraneo [24] proposed that people with higher education have more resources, such as cognition, communication, and interpersonal relationships, which enable them to make more informed choices and utilize more medical services.

In general, there are many studies on medical service utilization from the perspective of medical system, family, and individual factors, but there is scant research on the impact of social capital systematically. The social capital can be divided into two dimensions, including cognitive social capital and structural social capital [25]. Some scholars have, respectively, studied the impact of these two dimensions of social capital.

Research on cognitive social capital is mainly focused on the level of trust. Trust in others has a significant positive effect on health care utilization. An important way to improve the well-being of older adults in terms of health care utilization is to increase social capital, the most important of which is the norm of reciprocity and trust between individuals [26].

Research on structural social capital is mainly focused on social engagement and social networks. Social engagement can increase the likelihood of health care utilization [27]. A statistical study based on a large city in the United States found that residents with high levels of community social capital had easier access to health care resources [28]. The residents’ social participation can more or less contribute to more health-care-seeking behavior [29]. Chantal studied the impact of individual social capital on outpatient service utilization [30], and the results show that the more individuals live alone, the more important their social capital is likely to become. Besides social engagement, social network also affects health-care-utilization behavior. There are differences in health care utilization depending on the social network in which the individual is embedded [31]. Griffiths pointed out that the social network owned by individuals can provide them with more relevant information about medical institutions [32]. Bolin studied the relationship between individual structural social capital and the use of public health services by migrants [33].

From the above literature, social capital is an important factor affecting residents’ utilization of medical services. The scholars have studied the impact of related dimensions of social capital, including trust, social engagement, and social networks, but it is rare to systematically study social capital as a whole, and seldom does the literature test the impact of cognitive social capital and structural social capital on medical service utilization. From the perspective of research objects, there is much research on urban residents and the elderly, but scarcely does research focus on rural residents. In fact, compared with urban residents, social capital is more important for Chinese rural residents, whose interpersonal relations are tied closely by blood, kinship, and geography. Whether social capital affects the medical service utilization of rural residents needs to be tested empirically. Therefore, based on the data of China Family Panel Studies (CFPS) in 2018, this paper uses binary logit and multinomial logit models to empirically analyze the impact of social capital on rural residents’ medical service utilization in China. According to the results of empirical analysis, we put forward policy recommendations to provide theoretical guidance and decision-making references for improving rural residents’ medical service utilization and contributing to rural revitalization.

## 2. Variables and Hypothesis

### 2.1. Variables

#### 2.1.1. Medical Service Utilization

Medical service utilization refers to residents’ obtainment and enjoyment of professional medical services when they are sick. In 1982, the RAND Corporation proposed a two-part model approach to the measurement of medical service utilization. Then Duan systematically introduced this approach to the study of medical service, dividing medical service utilization into two phases, including “whether to seek medical treatment when sick” and “how much to spend on medical care” [34]. There is a consensus in taking “whether to seek medical treatment when sick” as the indicator for evaluating the first stage of medical service utilization. For the second stage, some scholars directly use the “cost of medical care” as a measure, while others use the “choice of medical institution” as an indicator when residents are sick [35].

It is debatable whether “cost of medical care” can fully and accurately reflect the utilization level of medical services. This is because the cost of medical care is largely influenced by the health status or type of disease of the individual, which vary from person to person. Some people are healthy, having no or minor illnesses, and spend less on medical care, while others are weak and even suffer from serious illnesses and spend more on medical care. Obviously, it cannot be concluded that people who are weak and spend more on medical care have better utilization of medical service than those who are healthy and spend less. Compared with “cost of medical care”, “choice of medical institution” can more objectively reflect the level of medical service utilization of residents. Especially in rural areas, the level of medical services varies greatly among different levels of medical institutions, such as clinics, village health offices, township health centers, specialty hospitals, and general hospitals. The higher the level of medical institutions are, the higher the level of medical services that medical institutions can provide. When rural residents are sick, the more opportunities they have to choose high-level medical institutions, which to a certain extent indicates that they make better use of good medical services. Therefore, for rural residents, it is appropriate to use the indicators of “whether to seek medical treatment when sick” and “choice of medical institution” to measure the utilization of medical services.

#### 2.1.2. Social Capital

The first systematic expression of the concept of social capital was made by the French sociologist Bourdieu, who believed that social capital is the aggregate of potential or actual resources possessed [36]. Coleman defined social capital as social structural resources owned by individuals, specifically manifested as information networks, obligations and expectations, authority relations, norms, and effective punishments [37]. Putnam’s definition of social capital is more community oriented. He believed that compared with human and physical capital, social capital emphasizes the inherent characteristics of social organizations such as norms, networks, and trust, which can promote social efficiency and increase the investment returns of human and physical capital [3].

Some scholars define social capital from the perspective of social network. Social capital refers to the ability of individuals to obtain resources in social networks, and this ability to obtain resources is an asset that can be used in social relations [38]. Social capital is the social resources in social activities for individual [39]. Its performance as a link between individuals or groups, including the social norms, social networks, and trust between the members of society [40]. The social participation can produce more medical treatment behavior [31,41].

In general, social capital should be divided into two dimensions, including cognitive social capital and structural capital [42]. Cognitive social capital refers to moral resources such as social trust, common norms, and values, while structural social capital refers to structural resources such as social networks and social participation.

Cognitive social capital is the capital connotation generated by guiding people to group action for common benefit based on the connotation of common norms, values, beliefs, and attitudes. Cognitive social capital reflects the thoughts and feelings in people’s hearts, so it is more subjective.

Structural social capital is the connotation of capital generated by promoting group actions for mutual benefit through the role and social network relationships established by normative rules, procedures, and precedents. The expression of structural social capital is relatively objective and more visible and can be constructed and developed through purposeful and conscious actions of groups [43].

### 2.2. Hypothesis

In China, social capital is an important part of social society especially in rural areas. Unlike urban “stranger society”, most rural residents have lived together for generations, knowing each other, trusting each other, and helping each other, thus accumulating rich social capital. Social capital of rural residents can have a positive impact on the utilization of medical services. The specific influence mechanism is as follows.

#### 2.2.1. Influence of Social Capital on “Whether to Seek Medical Treatment When Sick”

Whether rural residents seek medical treatment when they are sick mainly depends on their awareness and ability to seek medical treatment. The former shows whether rural residents think it is necessary to seek medical treatment, and the latter shows whether they could pay for medical treatment. Both can be positively influenced by social capital.

Firstly, social capital helps rural residents change their health perceptions and increase their awareness of health care. Richard found that participation in social organizations is an important form of social capital, and social organizations can influence members’ health perceptions and health behaviors by encouraging weight loss, prohibiting smoking, and urging members to get vaccinated [44]. This influence mechanism is also suitable for rural residents. In the countryside, especially in some remote and poor mountainous areas, due to the lack of correct health concepts, some villagers do not seek medical treatment when they are sick but rely on superstitious methods of treatment, causing mild diseases to evolve into serious diseases. If a rural resident joins a social organization or a social group or has more relatives and friends, when he is sick, the members of the association or relatives and friends can care about him and communicate with him about his illness and treatment methods; then, it will help him establish a correct health concept and realize the necessity of seeking medical treatment so that he will choose to seek medical treatment in time when he is sick.

Secondly, social capital is helpful for rural residents to make better use of medical policies and reduce the burden of medical expenses. Li analyzed the pro-wealth inequality and horizontal inequality in the utilization of medical services for the elderly using CHARLS data [45]. Disease, especially serious disease, is an important factor that leads to poverty among rural residents. Some villagers delay treatment and make the disease worse because they cannot afford medical expenses. Cebada found that the ill-health indicators contributed greatly to pro-poor inequality in the utilization of health services [46]. At the same time, low socioeconomic status not only worsened their health status but also significantly reduced their medical utilization [47]. In order to thoroughly solve the problem of poverty caused by illness and avoid the risk of returning to poverty due to illness, the Chinese government has introduced and implemented a series of health policies to help the poor. However, due to the overall low education level of rural residents and the lack of policy publicity in some places, some villagers do not understand the specific policies of health poverty alleviation and are not clear about which diseases can be treated for free, which items can be reimbursed, what percentage of reimbursement is available, and how to go through the relevant procedures, resulting in giving up the assistance they should have received. It is envisioned that if rural residents can learn about the relevant policies from their relatives, friends, and social organizations and receive guidance or assistance in utilizing the policies so that they can obtain medical services for free or at a lower out-of-pocket ratio, then rural residents will tend to choose to seek medical treatment in time when they are sick [48]. Based on this, the following hypothesis is proposed.

**Hypothesis** **1.***Good social capital promotes rural residents’ decision to seek treatment when they are sick*.

#### 2.2.2. Influence of Social Capital on Choice of Medical Institution

Social capital influences choice of medical institutions for rural residents in four main ways.

The first was is obtaining the convenience of transportation and accommodation. There are different levels of medical institutions, including clinics, village health offices, township health centers, specialty hospitals, and general hospitals. However, higher-level medical institutions are often located in county or urban areas, which are far away from the villages, and rural residents who go to see a doctor face the problems of poor transportation and off-site accommodation. For rural residents who do not have access to transportation and accommodation, they can only choose to seek medical treatment locally and nearby. On the contrary, if they have relatives and friends to help them solve the problems of transportation and accommodation for off-site medical treatment, they are more likely to choose to go to higher-level medical institutions.

The second way is receiving assistance of the procedures for medical treatment in a different location. Unlike the convenience of local medical treatment, off-site medical treatment requires referral and reimbursement procedures, which some rural residents with low education levels do not know how to handle, especially for vulnerable groups such as widows, orphans, and left-behind children. If friends and relatives share their experience of off-site medical treatment and help them with the cumbersome referral and reimbursement procedures, rural residents’ access to quality medical services can be increased.

The third way is broadening medical information channels. Unlike clinics and village health offices, the higher the level of hospital, the more detailed the division of departments, and for many rural residents, it is not clear which department they should choose to go to and which doctor’s number they should register. If relatives and friends help them know the information of high-level medical institutions, analyze the hospitals and departments suitable for their diseases, and recommend suitable doctor specialists, it is obvious that it will help rural residents choose high-level medical institutions.

The fourth way is reducing the cost of medical treatment [49]. Due to the information asymmetry between doctors and patients, doctors are at an information advantage, and patients are at an information disadvantage. Therefore, doctors may use their information advantage to induce demand of patient, such as repeated examinations, over-prescription, and over-treatment, which is called the moral hazard of doctors. Compared with local clinics, the information asymmetry between doctors and patients is stronger in an off-site hospital, and the moral hazard of doctors may be more serious. Some research findings show that patients with relationship capital can reduce unnecessary examinations, avoid unnecessary drugs and surgical procedures, and thus gain benefits such as “convenience” and “cost savings” of medical treatment [50].

In summary, the following hypothesis is proposed.

**Hypothesis** **2.***Good social capital promotes rural residents to choose high-level medical institutions when seeking medical treatment*.

## 3. Materials and Methods

### 3.1. Data

The data of this study were obtained from China Family Panel Studies (CFPS) in 2018, which was implemented by the China Social Science Survey Center of Peking University, with a sample covering 25 provinces/municipalities/autonomous regions, including all members of 16,000 households and a total of 32,669 valid questionnaires.

The target population of this paper is rural residents, and the research content is about the influence of social capital on the utilization of medical services. Therefore, the first step is to select suitable samples according to two criteria. One criterion is that “hukou is agriculture”, which means a rural household; the other criterion is that “had health problems in the past two weeks”, which means the patient. Thus, 6079 samples were obtained to analyze the influence of social capital on “whether to seek medical treatment when sick”, and the aim was to test Hypothesis 1. Then, we excluded the samples of those who did not seek medical treatment when they were sick, and a total of 4794 samples were obtained to further analyze the influence of social capital on the choice of medical institution to test Hypothesis 2.

### 3.2. Measurements

#### 3.2.1. Dependent Variable

The dependent variable is medical service utilization, including “whether to seek medical treatment when sick” and choice of medical institution. To measure the former, we selected the samples who had health problems in the past two weeks and asked them “have you seen a doctor?” If the answer was “Yes”, it was scored 1; otherwise, it was scored 0. About choice of medical institution, for the samples who had seen a doctor, we asked them “what type of medical institutions do you usually select if you see a doctor?” There are five options, including clinics, village health offices, township health centers, specialty hospitals, and general hospitals, and each answer was scored from 1 to 5 in turn.

#### 3.2.2. Independent Variable

The independent variable is social capital, including cognitive social capital and structural social capital. Cognitive social capital refers to moral resources such as social trust, shared norms, and values. Among them, social trust is divided into general trust and specific trust [51,52]. General trust reflects people’s perceptions of trust in the overall social environment. The question in the questionnaire is “Do you think most people can be trusted, or you should be more careful with people?” For those who selected the former, “most people can be trusted”, it was scored 1. For those who selected the latter, “should be more careful with people”, it was scored 0. Special trust reflects people’s trust in their acquaintances, which is generally measured by people’s trust in their neighbors. The question to measure is “How much do you trust your neighbors?” Scores were assigned from low to high on a scale of 0–10.

Shared norms and values are measured through religious beliefs. Many religions teach that one cannot live solely for one’s own selfishness but must learn to serve others voluntarily, and the premise of serving others is that you are healthy. That has become the prevailing ethic of religious organizations. Therefore, it is reasonable to assume that having religious beliefs can promote individuals’ concern for their own health to some extent. If they are sick, they go to see doctors to obtain treatment actively. Thus, religious beliefs positively influence the utilization of medical services. In order to determine whether the village residents have religious beliefs, we asked the question “Do you believe in Buddha or Bodhisattva?”, “Do you believe in God?”, “Do you believe in Allah?”, and “Do you believe in Jesus Christ?” It was scored 1 if the respondent chose “yes” for any of the above questions, indicating that the respondent had religious beliefs. It was scored 0 if the respondent chose “no” for all the above questions, indicating that the respondent had no religious beliefs.

Structural social capital refers to structural resources such as social networks and social participation. Among them, social network is measured by expenditure of family monetary gift. Since ancient times, Chinese people have been concerned with human relations, and a monetary gift is an important medium for building and maintaining interpersonal networks, especially for rural residents, who often need help from neighbors and relatives in production and living. It is very important for rural residents to develop and maintain their social networks through monetary gift. The question to measure was “How much did your family spend for monetary gift in the past 12 months?” Considering that this paper studies the social capital owned by individuals, the total expenditure of family monetary gift was divided by the numbers of family to obtain the per capita monetary gift expenditure. When processing data, the per capita expenditure of monetary gift was processed as logarithms. Social participation was measured by asking respondents “whether they are members of social organizations” [53], and the specific questions were “whether they are members of religious groups”, “whether they are members of the Communist Party of China“, “whether a member of a trade union”, and “whether a member of a self-employed workers’ association”. It was scored 1 if the answer was “yes” for any of these questions. It was scored 0 if the respondent chose “no” for all these questions.

#### 3.2.3. Control Variables

According to Behavioral Model of Health Services Use (BMHSU) founded by Andersen, this paper selected propensity factors (age, gender, marital status, education, whether smoked in the past month, and whether drank alcohol more than three times a week in the past month), need factors (health status, severity of injury or illness, and whether had chronic disease in the past six months), and ability factors (participation in medical insurance, per capita income) as control variables to ensure the scientific accuracy and comprehensiveness. Age is measured in years. Gender and marital status are scored as binary variables (1—male, 0—female; 1—married, 0—unmarried). Education level is divided into four scales ranged from 1—primary school and below, 2—middle school, 3—high school/technical school/vocational high school, to 4—undergraduate and above. Smoke, drink, and chronic disease are scored as binary variables (1—yes, 0—no). Health status is measured on a 5-point scale (1—not healthy, 2—relatively healthy, 3—healthy, 4—healthier, 5—very healthy). Severity of injury or illness is measured on a 3-point scale (1—not serious, 2—average, 3—serious). Participation in medical insurance is scored as binary variables (0—no, 1—yes), and the types of medical insurance include public medical care, urban employment insurance, urban residential insurance, supplemental medical insurance, and new rural cooperative medical insurance. Per capita income equals family income divided by numbers of family.

### 3.3. Variables Description

The names, measurement indicators, and scores of variables and descriptive statistics results are shown in Table 1. In order to fully reflect the characteristics of all samples, the samples in Hypotheses 1 and 2 are presented separately. The former 6079 samples are those rural residents who had health problems in the past two weeks. Excluding those who did not go to see a doctor, the latter 4794 samples are those who sought medical treatment when they were sick.

In terms of medical service utilization, 78.9% of rural residents choose to seek medical treatment when they are sick. While the mean value of medical institution is only 3.097, indicating that most rural patients seek medical institutions at the level of township health centers and below seldom go to higher-level institutions such as specialty hospitals and general hospitals.

Taking the former group as an example, the mean value of each dimension is as follows. The mean value of general trust in cognitive social capital is 0.489, indicating that rural residents’ trust in the social environment is at a moderately low level. The mean value of special trust is 6.559, which means the trust in neighbors is relatively high. The mean value of religious belief is 0.476, indicating that nearly half of the rural residents have religious beliefs. The average score of social participation in structural social capital is low, with only 11.2% of rural residents being members of social organizations. The mean value of per capita expenditure of monetary gift for the proxy variable of social network is up to CNY 1157.307 per year, accounting for 8.329% of per capita income (CNY 13,894.5). However, the standard deviation is large (2179.548), indicating that in terms of per capita expenditure of monetary gift, there are big differences among samples.

Regarding control variables, the average age of rural residents who had health problems in the past two weeks is 51.807 years, and the average age of those who chose to seek medical treatment is 53.527 years, indicating that the group with greater demand for medical service are mainly the middle-aged and elderly. Education is generally low, with a mean value of 1.737, indicating that for most, the education level is at middle school or below. Health status is generally poor, with a mean value of only 2.205, and the severity of injury or illness is relatively serious, with a mean value of 2.255. In addition, 31.9% of the samples suffer from chronic diseases. Rural residents who chose to seek medical treatment have weaker health status, with a mean value of only 2.098. Both the severity of injury or illness and chronic diseases are more serious, with mean values of 2.342 and 37.6% respectively. The medical insurance participation rate is high, up to 93.0%, which means that the construction of the Chinese medical security system in rural areas has achieved remarkable results.

## 4. Results

### 4.1. Social Capital and “Whether to Seek Medical Treatment When Sick”

The explained variable “whether to seek medical treatment when sick” is a dichotomous variable, so it is suitable for analysis by using a binary logit model. If rural residents did not seek medical treatment when they were sick, $y_{i}=0$; otherwise, $y_{i}=1$. The conditional probability that a rural resident *i* seeks medical treatment when sick is $P\left(y_{i}=1|x_{i}\right)=p_{i\;}$*(*i=1, 2⋯⋯n); then, $1-P_{i}$ is the probability that a rural resident does not seek medical treatment when he or she is sick, and the functional formula is shown in (1).
(1)$$p_{i}=\frac{exp\left(\alpha +\beta x_{i}\right)}{1+exp\left(\alpha +\beta x_{i}\right)}$$

Further translating this into a logarithmic formula,
(2)$$\ln\frac{p_{i}}{1-p_{i}}=\alpha +\beta x_{i}+\varepsilon_{i}$$

In the above formula, $\ln\frac{p_{i}}{1-p_{i}}$ is logarithmic odds ratio. Among them, $\frac{P_{i}}{1-P_{i}}$ means that the probability of individuals choosing to see a doctor when they are sick is $\frac{p_{i}}{1-p_{i}}$ times the probability of not seeing a doctor. The regression coefficient β reflects the degree and direction of the influence of each factor (social capital and control variables) on whether rural residents seek medical treatment.

Model 1 uses only control variables, and model 2 adds social capital variables into model 1 (shown in Table 2). Regression results are reported in Table 2 for both regression coefficients β and power value $Exp\left(\beta \right)$. The power value $Exp\left(\beta \right)$ reflects the change in the probability of rural residents choosing to seek medical treatment when they are sick for each unit change in the independent variable. The results show that the corresponding regression coefficients β and powers $Exp\left(\beta \right)$ do not change significantly before and after putting in social capital variables, indicating to a certain extent that the model is stable.

Among the control variables, age, marital status, severity of injury or illness, chronic disease, and participation in medical insurance have positive influence, but education, drinking alcohol, and health status have negative influence on whether to seek medical treatment. Other variables’ influence does not pass the significance test. The influences of cognitive social capital and structural social capital on whether to seek medical treatment are as follows.

#### 4.1.1. Influence of Cognitive Social Capital on “Whether to Seek Medical Treatment”

The empirical results show that special trust and religious beliefs have a significant positive effect on rural residents’ choice to seek medical treatment when they are sick, and the coefficients of special trust and religious beliefs are 0.031 and 0.122, respectively. As shown in column (4), holding other variables constant, if special trust, i.e., trust in neighbors, increases by one point, the probability of rural residents choosing to seek medical treatment when sick increases by 3.2% (*p* < 0.05). The reason for this is that the more rural residents trust their neighbors; the more willing they are to communicate with neighbors about their illness, the more their neighbors know their illness and can remind them to seek medical treatment in time. Compared to those without religious beliefs, rural residents with religious beliefs were 13.0% more likely to seek medical treatment when they were sick (*p* < 0.1). The reason may be that religious beliefs can influence the value of life for religious people. Many religious teachings advocate people to cherish life and establish a correct view of health, which undoubtedly makes the religious people pay more attention to their health and not delay medical treatment when they are sick.

The lack of significant influence of general trust on whether to seek medical treatment may be related to the “differential pattern” of social relations in the Chinese countryside. The “differential pattern” was proposed by Mr. Fei Xiaotong, which occurs in social relationships such as kinship and geopolitical relationships. With “self” as the center, like a stone thrown into water, the water ripples out in circles, forming a social relationship structure that is pushed further and further away and becomes thinner and thinner. According to the theory of “differential pattern”, the larger the radius of the differential pattern, the more distant the relationship is and the lower the level of trust. Compared with neighbors, rural residents lack sufficient trust in groups with a larger radius of interaction, such as government agencies, social organizations, and strangers, etc. The results in Table 1 also confirm that only 48.9%, i.e., less than half, of rural residents believe that most people can be trusted. Due to the low level of general trust, rural residents would not inform people with more distant relationships about their illnesses, and therefore, it is difficult for general trust to have an impact on whether rural residents seek medical treatment.

#### 4.1.2. Influence of Structural Social Capital on “Whether to Seek Medical Treatment”

The empirical results show that neither social network nor social participation has a significant effect on whether rural residents seek medical treatment when they are sick, and the reasons for this are analyzed below.

As a proxy variable for social networks, the amount of per capita expenditure of monetary gift reflects the interpersonal resources that rural residents have. However, in some villages, the culture of comparison has led to an increase in monetary gift expenses and has even become a heavy financial burden for rural residents.

In the countryside, there is a wide range of monetary gift expenses, such as wedding, funeral, promotion, birth, birthday, housewarming, company opening, etc., and villagers have a serious view of “face”, considering that the more the people attend their banquet, the more they have “face”. Therefore, the scope of invitations has become wider and wider, from relatives to the same surname and even poker friends. In fact, some of the monetary gift correspondents are not close to them, and some of them are even very distant. Then, when they are sick, they do not communicate with each other, and the monetary gift correspondent does not know about their illness and cannot help them to make use of medical services. Hence, the expenditure of monetary gift has no effect on whether rural residents seek medical treatment when they are sick.

Being a member of a social organization as a proxy variable for social participation does not significantly affect rural residents’ decision to seek medical treatment when they are sick, and the main reason may be that rural residents who are members of social organizations are required to undertake more tasks beyond their own job. They are busy and have no time to see a doctor, so they may not seek medical treatment when their illness is not very serious.

### 4.2. Social Capital and Choice of Medical Institution

The explained variable choice of medical institution is suitable for multinomial choice model and is thus subjected to regression analysis by using the multinomial logit model. The multinomial logit model can be considered as an estimation of multiple binary logit models implemented after two-by-two matching of different types of medical institutions.

The choice of medical institution is as $\omega_{i}$ and takes values from 1 to 5, representing clinics, village health offices, township health centers, specialty hospitals, and general hospitals. The probability of rural residents choosing a particular type of medical institution when seeking medical treatment is $P\left(\omega_{i}=j|x_{i}\right)=p_{ij\;}\left(i=1,2\cdots \cdots n;\;j=1,2\cdots \cdots 5\right)$. The functional form is shown in (3).
(3)$$P\left(\omega_{i}=j|x_{i}\right)=\frac{exp\left(\alpha_{j}+\beta_{j}x_{i}\right)}{\sum_{k=1}^{j}exp\left(\alpha_{j}+\beta_{k}x_{i}\right)}$$

Further, we analyzed the probability of choosing another type of medical institution compared to the clinic by taking the clinic ($\omega_{i}=1)$ as the “reference group” and making its coefficient $\beta_{1}$ = 0. The probability of an individual i choosing the option j is
(4)$$P\left(\omega_{i}=j|x_{i}\right)=\left\{\begin{array}{c} \frac{1}{1+\sum_{k=2}^{j}exp\left(\alpha_{j}+\beta_{k}x_{i}\right)}\left(j=1\right)\\ \frac{exp\left(\alpha_{j}+\beta_{k}x_{i}\right)}{1+\sum_{k=2}^{J}exp\left(\alpha_{j}+\beta_{k}x_{i}\right)}\left(j=2,3\cdots J\right) \end{array}\right.$$

Converting the above formula into a logarithmic expression,
(5)$$\ln\left(\frac{P\left(\omega_{i}=j\left(j\geq 2\right)|x_{i}\right)}{P\left(\omega_{i}=1|x_{i}\right)}\right)=\alpha_{j}+\beta_{j}x_{i}+\varepsilon_{ij}\left(j=2,3\cdots J\right)$$

$\beta_{j}$ reflects the degree and direction of influence of each factor (social capital and control variables) on rural residents’ choice of medical institution. The regression results report regression coefficients $\beta_{j}$ and power value $\;Exp\left(\beta_{j}\right)$. The power value $Exp\left(\beta_{j}\right)$ reflects the multiplicative change in the probability of causing rural residents to choose other type of medical institutions relative to choosing a clinic for each small amount of change in $\;x_{i}$. The regression results are shown in Table 3.

None of the control variables has significant impact on all the choice types of medical institution, but some control variables significantly influence some types of choice. For example, age, gender, education, severity of injury or illness, chronic disease, participation in medical insurance, and per capita income have positive influence, but smoking and drink alcohol have negative influence on some of the choice types. Marital status and health status have no influence, failing to pass the significance test. The influences of cognitive social capital and structural social capital on choice of medical institution are as follows.

#### 4.2.1. Influence of Cognitive Social Capital on Choice of Medical Institution

The regression results in Table 3 show that none of the three elements of cognitive social capital (universal trust, special trust, and religious belief) have a significant effect on choice of medical institution.

The insignificant effect of general trust may be due to the shaking of the “acquaintance society” structure in villages. In some villages, the practice of “village conversion” has been carried out through “village evacuation” and “farmers moving to buildings”. Additionally, the way villagers live has changed from “horizontally scattered” to “vertically concentrated”. Under the “horizontally scattered” residence mode, it is convenient for villagers to visit and communicate with neighbors, which is helpful for mutual trust. While under the “vertically concentrated” residence mode, the buildings hinder communication and interaction among villagers, and interpersonal relationships become distant, and the general trust is weakened.

Special trust (i.e., trust in neighbors) does not significantly affect choice of medical institution, which may be related to the low literacy level of the samples and their neighbors in the rural areas. Choosing a medical institution requires a better understanding of information about disease types, departments, and specialists and requires a higher level of literacy. However, Table 1 shows that the average education score of all samples was only 1.737, indicating that most rural residents’ education is at the middle school level or below. Because of the overall low education level and limited ability, the neighbors could not provide effective help to rural residents in choosing a medical institution.

Religious beliefs influence whether to seek medical treatment but have no significant effect on choice of medical institution. The main reason is that religious beliefs have a “belief effect” rather than an “organizational effect” on medical treatment. The belief effect is that religion influences individuals’ life values and promotes them to value their life and health so that they positively seek medical treatment when they are sick. The organization effect is that religion influences individuals’ behavior through collective activities, such as medical lectures, health consultations, and charity clinics, which help individuals to choose medical institutions. However, rural religions in China are less organized and in a looser state and have not yet formed an effective information sharing and resource support system, which does not provide strong medical support to the religious patients.

#### 4.2.2. Influence of Structural Social Capital on Choice of Medical Institution

Social networks have a significant positive effect on the choice of township health centers, specialty hospitals, and general hospitals except for the effect on village health offices, which does not pass the significance test. The effects on the choice of specialty and general hospitals are comparable, and for each unit increase in per capita expenditure of monetary gift, the probability of rural residents choosing specialty and general hospitals increased by 7.3% and 7.0%, respectively, over the probability of choosing clinics, both of which are higher than the effect on the choice of township health centers (5.4%). This result is contrasted with the lack of effect of social networks on whether to seek medical treatment in Table 2.

Why does the proxy variable of social network, the expenditure of monetary gift, have no effect on whether to seek medical treatment but influences the choice of medical institution? This may be related to the closeness of social relationships and the differences in medical decision making at different stages of medical treatment. Rural residents have a wide range of recipients for monetary gift spending, and some are even distantly related. For the decision of whether to seek medical treatment when sick, rural residents usually only discuss with their family members or those who are particularly close to them and do not inform most of the recipients of monetary gift spending. Therefore, expenditure of monetary gift does not have an impact on their decision of whether to seek medical treatment. However, at the stage of deciding to seek medical treatment and choosing a medical institution, due to the lack of information about medical institutions in other places, the rural patients need help from social networks to solve the problems of information asymmetry, and the higher the level of medical institutions, the more serious the information asymmetry is. At this stage, it is necessary for rural patients to mobilize a wider range of interpersonal resources to consult information and seek medical help; thus, the social network has an active role.

Social participation has a significant positive effect on choice of medical institution. The probability ratios of choosing general hospitals, specialized hospitals, township health centers, village health offices, clinics are 46.6%, 15.2%, 38.1%, and 41.3% higher, respectively, for rural residents who participate in social organizations compared to non-members of social organizations. In other words, participation in social organizations helps rural residents to choose medical institutions with more medical resources and better medical service capacity. The main reason is that social organizations provide a platform for members to share information and promote mutual help and assistance. On one hand, compared with individuals, social organizations have an advantage in information collection, and members can obtain information about medical institutions through the relevant platforms of social organizations. On the other hand, common organizational affiliation gives members a strong sense of trust, and when a member has a medical need, other members are willing to provide support or help, thus increasing the probability that patients can obtain high-level medical services.

## 5. Conclusions and Suggestions

This paper empirically analyzes the influence of social capital on rural residents’ “whether to seek medical treatment when sick” and “choice of medical institution” based on the 2018 CFPS data using binary logit and multinomial models. The results show that among the cognitive social capital, apart from general trust, special trust and religious beliefs have a significant positive influence on whether rural residents choose to seek medical treatment. Among the structural social capital, social participation has a significant positive influence on choice of medical institution, and social network has a significant positive effect on the choice of township health centers, specialty hospitals, and general hospitals, but the influence on village health centers did not pass the significance test. Therefore, Hypotheses 1 and 2 are partially tested.

Overall, social capital positively contributes to rural residents’ medical service utilization, and the effects of cognitive social capital and structural social capital differ at different stages of medical service utilization. In the stage of deciding whether to seek medical treatment when rural residents are sick, the main influencing factor is cognitive social capital; while in the stage of choosing medical institutions, structural social capital plays a major role. Based on the empirical results, this paper proposes the following suggestions.

### 5.1. Reconstruct the Trust Mechanism in Rural Society

General trust has no effect on the rural residents’ utilization of medical services, and special trust only has a weak effect on “whether to seek medical treatment”. This reflects the problem of loss of social trust capital in the countryside. “Being good to neighbors and partnering with neighbors” is the traditional virtue of the Chinese nation, which has a good cultural foundation especially in rural areas. This needs to be inherited and carried forward through collective activities and mutual help in the community to promote villagers’ communication and interaction and thus enhance mutual trust.

### 5.2. Play the Positive Role of Religious Beliefs

The empirical results show that religious beliefs have a positive effect on whether to seek medical treatment, mainly because of the “faith effect”, which enables individuals to establish correct values of life. However, the “organizational effect” of religious beliefs is weak and does not have a significant effect on choice of medical institution. Therefore, it is necessary to further develop the positive value guidance function of religion in medical treatment so that religious people can value life and health, treat diseases scientifically, and strengthen their awareness of medical treatment. Furthermore, it is necessary to improve the rural religious management system, change its loose status, and organize activities such as health consultation, medical insurance policy propaganda, and training to enhance the ability of the religious people to seek medical treatment.

### 5.3. Promote the Healthy Development of Social Networks

The social network of rural residents, with expenditure of monetary gift as a proxy variable, has no significant effect on whether to seek medical treatment. In some villages, gifts have become a kind of “face culture”, with too many items and high amounts, which have become a heavy financial burden for rural residents. Therefore, it is necessary to strengthen the construction of village rules and regulations through democratic consultation so as to eliminate the wind of wastefulness and comparison and reshape the healthy ecology of rural residents’ relationships.

### 5.4. Foster and Develop Rural Social Organizations

The empirical results show that rural residents’ participation in social organizations significantly increases their probability of obtaining high-quality medical services. Social organizations can enhance residents’ sense of belonging and trust in each other, and they play an important role in the rural governance. However, there are few professional social organizations in villages, and especially, medical and health social organizations are still lacking. It is vital to foster rural autonomous organizations and support the expansion of urban social organizations into the countryside, helping the rural residents to make better use of medical services, thus strengthening rural health human capital and contributing to rural revitalization.

## Figures and Tables

**Table 1 ijerph-19-15989-t001:** Variables and descriptive statistics results.

Variable Names		Measurement Indicators	Variable Scores	Hypothesis 1 (*N* = 6079)		Hypothesis 2 (*N* = 4794)	
				Mean Value	Standard Deviation	Mean Value	Standard Deviation
Dependent Variable	Medical Service Utilization	Whether to seek medical treatment when you are sick	0—No, 1—Yes	0.789	0.408	-	-
		Choice of medical institution	1—Clinic 2—Village Health Office 3—Township Health Center 4—Specialty Hospital 5—General Hospital	-	-	3.097	1.540
Independent Variable	Cognitive Social Capital	General trust	0—Be more careful with people 1—Most people can be trusted	0.489	0.500	0.488	0.500
		Special trust	Score 0–10	6.559	2.270	6.598	2.303
		Religious beliefs	0—No, 1—Yes	0.476	0.499	0.488	0.500
	Structural Social Capital	Social network	Per capita expenditure of family monetary gift	1157.307	2179.548	1115.79	1707.079
		Social participation	0—No, 1—Yes	0.112	0.315	0.109	0.312
Control variables	Propensity Factors	Age	Actual years	51.807	15.320	53.527	14.642
		Gender	0—Female, 1—Male	0.399	0.490	0.384	0.486
		Marital status	0—Unmarried, 1—Married	0.835	0.372	0.853	0.354
		Education	1—Primary school and below 2—Middle school 3—High school/Technical school/Vocational high school 4—Undergraduate and above	1.737	0.701	1.676	0.669
		Did you smoke in the past month	0—No, 1—Yes	0.258	0.438	0.247	0.431
		Did you drink alcohol more than three times a week in the past month	0—No, 1—Yes	0.124	0.329	0.112	0.315
	Needs factors	Health status	1—Not healthy 2—Relatively healthy 3—Healthy 4—Healthier 5—Very healthy	2.205	1.192	2.098	1.183
		Severity of injury or illness	1—Not serious 2—Average 3—Serious	2.255	0.719	2.342	0.701
		Chronic disease in the past six months	0—No, 1—Yes	0.319	0.466	0.376	0.485
	Ability factors	Participation in medical insurance	0—No 1—Yes (including: public medical care/urban employment insurance/urban residential insurance/supplemental medical insurance/New Rural Cooperative Medical Insurance)	0.930	0.255	0.937	0.243
		Per capita income	Family income divided by numbers of family	13,894.5	21,049.47	13,160	19,458.62

**Table 2 ijerph-19-15989-t002:** Regression results for social capital and “whether to seek medical treatment” (*N* = 6079).

	Model 1		Model 2	
	(1)	(2)	(3)	(4)
	β	$Exp\left(\beta \right)$	β	$Exp\left(\beta \right)$
Age	0.017 *** (6.645)	1.017 *** (6.645)	0.017 *** (6.207)	1.017 *** (6.207)
Gender	0.001 (0.012)	1.001 (0.012)	0.052 (0.565)	1.054 (0.565)
Marital status	0.232 *** (2.758)	1.262 *** (2.758)	0.256 *** (2.937)	1.291 *** (2.937)
Education	−0.154 *** (−2.846)	0.857 *** (−2.846)	−0.139 ** (−2.446)	0.870 ** (−2.446)
Did you smoke in the past month	−0.089 (−0.948)	0.915 (−0.948)	−0.138 (−1.408)	0.871 (−1.408)
Did you drink alcohol more than three times a week in the past month	−0.477 *** (−4.895)	0.621 *** (−4.895)	−0.465 *** (−4.556)	0.628 *** (−4.556)
Health status	−0.063 ** (−2.169)	0.939 ** (−2.169)	−0.069 ** (−2.266)	0.933 ** (−2.266)
Severity of injury or illness	0.529 *** (11.219)	1.698 *** (11.219)	0.542 *** (11.111)	1.719 *** (11.111)
Chronic disease in the past six months	1.246 *** (12.906)	3.475 *** (12.906)	1.257 *** (12.593)	3.514 *** (12.593)
Participation in medical insurance	0.287 ** (2.374)	1.332 ** (2.374)	0.280 ** (2.228)	1.323 ** (2.228)
Per capita income	−0.010 (−0.415)	0.990 (−0.415)	−0.015 (−0.573)	0.985 (−0.573)
General trust	-	-	0.043 (0.597)	1.043 (0.597)
Special trust	-	-	0.031 ** (1.993)	1.032 ** (1.993)
Religious beliefs	-	-	0.122 * (1.763)	1.130 * (1.763)
Social network	-	-	0.026 (1.614)	1.026 (1.614)
Social participation	-	-	−0.135 (−1.276)	0.874 (−1.276)

Note: z-values are in parentheses; *** *p* < 0.01, ** *p* < 0.05, * *p* < 0.1.

**Table 3 ijerph-19-15989-t003:** Regression results for social capital and choice of medical institution (*N* = 4794).

	General Hospitals		Specialized Hospitals		Township Health Centers		Village Health Offices	
	(1)	(2)	(3)	(4)	(5)	(6)	(7)	(8)
	β	$Exp\left(\beta \right)\;$	β	$Exp\left(\beta \right)\;$	β	$Exp\left(\beta \right)\;$	β	$Exp\left(\beta \right)\;$
Age	0.017 *** (5.138)	1.017 *** (5.138)	−0.002 (−0.333)	0.998 (−0.333)	0.029 *** (8.078)	1.029 *** (8.078)	0.033 *** (8.202)	1.033 *** (8.202)
Gender	0.229 ** (2.022)	1.257 ** (2.022)	0.571 *** (3.082)	1.770 *** (3.082)	0.257 ** (2.157)	1.293 ** (2.157)	−0.007 (−0.049)	0.993 (−0.049)
Marital status	−0.006 (−0.053)	0.994 (−0.053)	0.254 (1.182)	1.290 (1.182)	0.087 (0.702)	1.091 (0.702)	−0.083 (−0.619)	0.920 (−0.619)
Education	0.204 *** (2.847)	1.226 *** (2.847)	0.205 * (1.661)	1.227 * (1.661)	0.039 (0.513)	1.040 (0.513)	0.153 * (1.794)	1.166 * (1.794)
Did you smoke in the past month	−0.275 ** (−2.249)	0.760 ** (−2.249)	−0.520 ** (−2.514)	0.595 ** (−2.514)	−0.257 ** (−2.014)	0.773 ** (−2.014)	0.079 (0.551)	1.082 (0.551)
Did you drink alcohol more than three times a week in the past month	−0.312 ** (−2.228)	0.732 ** (−2.228)	−0.224 (−0.922)	0.799 (−0.922)	−0.091 (−0.643)	0.913 (−0.643)	−0.063 (−0.401)	0.939 (−0.401)
Health status	0.008 (0.216)	1.008 (0.216)	−0.050 (−0.736)	0.951 (−0.736)	0.040 (1.014)	1.041 (1.014)	0.055 (1.248)	1.057 (1.248)
Severity of injury or illness	0.324 *** (5.163)	1.383 *** (5.163)	0.519 *** (4.423)	1.681 *** (4.423)	0.030 (0.468)	1.031 (0.468)	−0.102 (−1.423)	0.903 (−1.423)
Chronic disease in the past six months	0.428 *** (4.827)	1.535 *** (4.827)	0.326 ** (2.142)	1.386 ** (2.142)	0.043 (0.454)	1.044 (0.454)	−0.110 (−1.022)	0.895 (−1.022)
Participation in medical insurance	0.246 (1.562)	1.279 (1.562)	0.397 (1.321)	1.487 (1.321)	0.622 *** (3.354)	1.863 *** (3.354)	0.092 (0.504)	1.097 (0.504)
Per capita income	0.062 * (1.895)	1.064 * (1.895)	−0.070 (−1.457)	0.932 (−1.457)	0.003 (0.078)	1.003 (0.078)	0.001 (0.020)	1.001 (0.020)
General trust	0.139 (1.640)	1.150 (1.640)	0.071 (0.477)	1.073 (0.477)	0.137 (1.527)	1.147 (1.527)	0.016 (0.161)	1.016 (0.161)
Special trust	−0.006 (−0.312)	0.994 (−0.312)	−0.021 (−0.648)	0.979 (−0.648)	−0.000 (−0.023)	1.000 (−0.023)	0.019 (0.858)	1.019 (0.858)
Religious beliefs	0.003 (0.032)	1.003 (0.032)	0.030 (0.208)	1.031 (0.208)	−0.150 (−1.706)	0.860 (−1.706)	−0.094 (−0.950)	0.910 (−0.950)
Social network	0.068 *** (3.386)	1.070 *** (3.386)	0.070 * (1.931)	1.073 * (1.931)	0.053 ** (2.530)	1.054 ** (2.530)	0.022 (0.976)	1.022 (0.976)
Social participation	0.383 *** (2.764)	1.466 *** (2.764)	0.142 * (0.578)	1.152 * (0.578)	0.323 ** (2.201)	1.381 ** (2.201)	0.346 ** (2.152)	1.413 ** (2.152)

Note: z-values are in parentheses; *** *p* < 0.01, ** *p* < 0.05, * *p* < 0.1.

## Data Availability

Not applicable.

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
