# Peer review of "The Impact of Social Capital on Rural Residents’ Medical Service Utilization in China—An Empirical Study Based on CFPS Data"

_ijerph, 2022, doi:10.3390/ijerph192315989_

Round 1
Reviewer 1 Report
It is a very important topic to discuss the impact of social capital on rural residents' medical service utilization. However, there are some major problems.
First, when measuring social capital, you consider cognitive social capital and structural social capital. However, the measurement and measurement of these two types of social capital are controversial, and you need to explain this. In addition, from the process of estimating results, it is impossible to identify which type of social capital is more important because you use multiple indicator agents.
Second, the article only uses the 2018 CFPS, which is not enough to support the research conclusion.
Third, the topic discussed in this article is the impact of social capital on the utilization of medical services by rural residents. I can't agree with the content of Section 4.1.2, because this secetion involves the comparison between urban and rural areas, which is not closely related to the research topic, which may easily lead to the lack of focus of the research content.
Author Response
Dear reviewer,
Thank you for your detailed review of my article (ijerph-2012359), your approval of the topic, and your practical suggestions regarding the literature review, data use, and empirical analysis. Based on your valuable and appropriate suggestions, I have revised the article and marked it for your review through the "Track Changes" feature. Below is my detailed response to your comments.
Point 1: when measuring social capital, you consider cognitive social capital and structural social capital. However, the measurement and measurement of these two types of social capital are controversial, and you need to explain this. In addition, from the process of estimating results, it is impossible to identify which type of social capital is more important because you use multiple indicator agents.
Response 1: Your question about multiple measures has caused me to revisit my article and has had a significant impact on my line of writing. For the measurement and measurement of social capital, since Bourdieu introduced it into the field of social science, Coleman, Uphoff, Adler, Brown, Linnan, Putnam, Granovetter, and others have categorized and measured social capital at the conceptual level, individual level, and collective level, respectively, and indeed different scholars have different ways of categorizing and measuring it. At present, the more authoritative and academically recognized research on the classification measurement of structural social capital and cognitive social capital is Uphoff, so the article uses Uphoff's classification of social capital as the standard and is annotated in the text. (25)
This article points out the importance of both cognitive and structural social capital, not to compare the relationship between which of the two has a more significant impact on rural residents' health care utilization, but to point out the impact of both when it comes to rural residents' health care utilization. The focus of the study is on the relationship between social capital and health care utilization of rural residents, not on the further delineation of which social capital has a stronger impact within social capital.
Point 2: the article only uses the 2018 CFPS, which is not enough to support the research conclusion.
Response 2:
Thank you for the authors' consideration of using only 2018 CFPS data for the study. Continuous multi-year data provide information on dynamic behavior and are suitable for studying changes in the same scope over time, mainly for time series analysis. The article examines the impact of social capital on health care utilization, and the key variables are social capital and health care utilization status; therefore, cross-sectional data are used, primarily to examine differences across individuals at the same time, and are not a selection to examine trends.
In addition, the article uses the CFPS data for 2018. Its full name is the Chinese Household Dynamics Tracking Survey. This project is a major social science project implemented by the China Social Science Survey Center of Peking University with the goal of providing data for academic research and policy decisions by tracking and collecting data at three levels: individual, household, and community, and responding to social, economic, demographic, educational, and health changes in China. The project has been implemented for 12 years with this fixed question, and the sample covers 25 provinces/municipalities/autonomous regions, with a target sample size of 16,000 households, and the survey subjects include all family members in the sample households, so its survey data have high reliability and stability, and are relatively convincing.
Point 3: the topic discussed in this article is the impact of social capital on the utilization of medical services by rural residents. I can't agree with the content of Section 4.1.2, because this secetion involves the comparison between urban and rural areas, which is not closely related to the research topic, which may easily lead to the lack of focus of the research content.
Response 3: Thank you for your comment on the article that "urban-rural comparisons tend to lead to unfocused research". The original purpose of this section of the article was to express whether structural social capital affects rural residents' access to health care, and the presentation of the proxy variable of social network in structural social capital did dilute the topic. To ensure the structural integrity of the entire article, we have therefore removed the table and only included an explanation in the text.
We have carefully studied your comments and made corrections accordingly. Please find attached the original manuscript revised according to the reviewer's comments, and the revised part is marked with the "track changes" function for your quick review, please see the attachment for the detailed revision: Revised version ijerph-2012359
Thank you again for your valuable comments on my article, and I hope to receive your approval.
Yours sincerely,
Corresponding author:Jiqing Hong
mail: hej96@stu.zuel.edu.cn

Reviewer 2 Report
Comments on The Impact of Social Capital on Rural Residents' Medical Ser-2 vice Utilization in China—An Empirical Study Based on CFPS Data
Generally good paper
Strengths of Paper
Appropriate choice of model
Important and timely topic
Interesting, sufficient data
Important bifurcated finding on social capital
Weaknesses of Paper
Drop first 12 lines of paper and start with “Equitable access and utilization..
Descriptive stats issues In Table 1—the mean value is not that helpful. It would be more useful to know what percent of the sample chose each type of medical institution
|
Choice of medical institution |
1-Clinic, 2-Village Health Office, 3-Township Health Center, 4-Specialty Hospi-tal, 5-General Hospital |
Too much explication of models—the binary and multinomial logits are well understood and do not need to be presented in such detail
Overstate confidence of motives of rural residents of results—social behaviors to explain non results should be couched more as conjecture because they are speculative. The evidence is not provided for a definitive, only a potential, determinant of what is observed in the data
Line 616 has a reference #50 with no corresponding entry in the references
Author Response
Dear reviewer,
Thank you for your detailed review of my article (ijerph-2012359), your approval of the topic, model, data, and theory, and your practical suggestions regarding the introduction, descriptive statistics, empirical analysis, and literature citations. Based on your valuable and appropriate suggestions, I have revised the article and marked it for your review through the "Track Changes" feature. Below is my detailed response to your comments.
Point 1: Drop first 12 lines of paper and start with“Equitable access and utilization.
Response 1: Your suggestion is very pertinent and opens the article on point. I have revised it according to your suggestion, please see the revised article for details.
Point 2: Descriptive stats issues In Table 1—the mean value is not that helpful. It would be more useful to know what percent of the sample chose each type of medical institution
Choice of medical institution.
1-Clinic, 2-Village Health Office, 3-Township Health Center, 4-Specialty Hospi-tal, 5-General Hospital
Response 2: It is reasonable of use percentages to show the choice of medical institution, and we have considered to use it before. However, we did not use them because of the following two reasons.
For one thing, the choice of medical institution is not a single choice, but multiple choices. Statics data show that some residents choose one medical institution, some choose several medical institutions because of referral by the doctors. So, the proportions overlap, and the proportions do not add up to 100%.
In addition,since all the other variables in the table is the mean value, in order to be consistent and coordinate of the whole table, it is better to use mean value for the choice types of medical institutions.
Point 3: Too much explication of models—the binary and multinomial logits are well understood and do not need to be presented in such detail.
Response 3: Thank you for your suggestions on the presentation of the model, which made it clear to me that the article is cumbersome. Based on your suggestion, I have dropped some of the simple descriptions of the binary and multinomial logarithms in the article, which can be found in the article. Some of the model descriptions have been retained in the article in order to make the model understandable to other professionals and to facilitate a quicker understanding of the article.
Point 4: Overstate confidence of motives of rural residents of results—social behaviors to explain non results should be couched more as conjecture because they are speculative. The evidence is not provided for a definitive, only a potential, determinant of what is observed in the data.
Response 4: What you said is quite reasonable. The data can objectively reflect the research conclusion, so data description and analysis are extremely important. After data analysis, in order to provide a solid foundation for countermeasure suggestions, we further explore its reasons and influence mechanism behind by literature and interview. Based on these, we proposed countermeasures to make it be more targeted and persuasive.
Point 5: Line 616 has a reference #50 with no corresponding entry in the references.
Response 5: Thank you very much for pointing out this reference error in our article. We have promptly confirmed and corrected it, and have removed the "reference 50".
We have carefully studied your comments and made corrections accordingly. Please find attached the original manuscript revised according to the reviewer's comments, and the revised part is marked with the "track changes" function for your quick review, please see the attachment for the detailed revision: Revised version ijerph-2012359
Thank you again for your valuable comments on my article, and I hope to receive your approval.
Yours sincerely,
Corresponding author:Jiqing Hong
E-mail: hej96@stu.zuel.edu.cn

Reviewer 3 Report
The article herein deals with an interesting and timely topic. However, I elaborate on my concerns below and hope that they help authors strengthen their contributions.
1. The authors state that “…..there are many studies on medical service utilization from the perspective of medical system, family, and individual factors, but only a few studies are conducted from the perspective of social capital, and seldom studies on rural residents.” How do findings on the influence of social capital on medical service utilization compare with the previous literature? Especially, Is the finding that structural social capital is effective in choosing medical institutions while cognitive social capital is important in whether to seek treatment validated by the extant literature?
2. How about the estimated coefficients of control variables (i.e., propensity factors)? How do they relate to the dependent variables? How do you evaluate the estimation results with regard to them? [522-523] & [433-434]
3. The authors’ choice of medical institution includes: “1-Clinic, 2-Village Health Office, 3-Township Health Center, 4-Specialty Hospital, 5-General Hospital”. However, estimation results as to the multinomial logit model in Table 4 as to the influence of social capital on the choice of medical institution do not report results with regard to the clinic. Are they missing? or is there any other reason for their non-exclusion? [371-372] & [522-523]
4. There are recurring questions designed to determine whether a given sample has religious beliefs: “Do you believe in God?” and it shall be removed accordingly. [320-326]
5. One of the criteria used to choose the sample is “had health problems in the past two weeks” which constitutes “patient”. What is the rationale behind the choice of the past two weeks? [282-291]
6. To my understanding, among the propensity factors- control variables- questions: “Did you smoke in the past month” and “Did you drink alcohol more than three times a week in the past month” are there to check whether a given sample has smoking and alcohol consumption habits? Would not it be better had the questions directly asked a given sample whether they smoke or consume alcohol on a regular basis? Smoking in the past month or drinking alcohol more than three times a week in the past month may not alone be able to lead to the need for medical service utilization on the part of a patient. [346-347]
7. How do authors evaluate the relatively low values of Pseudo R² in Table 2 and Table 4? [433-434] & [522-523]
8. The authors stated that the mean values as to variables included in the study are not statistically different [378-379]. Did the authors check whether statistically significant differences exist between included variables with respect to the two samples via any statistical test?
Author Response
Dear reviewer,
Thank you for your detailed review of my article (ijerph-2012359), your approval of the topic selection, and your practical suggestions regarding the literature review, data sources (sample selection criteria, control variable issues), and empirical analysis (control variable description, clinic analysis description, goodness-of-fit test, sample description). Based on your valuable and appropriate suggestions, I have revised the article and marked it with the "Track Changes" feature for your review. Below is my detailed response to your.
Point 1: The authors state that “…..there are many studies on medical service utilization from the perspective of medical system, family, and individual factors, but only a few studies are conducted from the perspective of social capital, and seldom studies on rural residents.” How do findings on the influence of social capital on medical service utilization compare with the previous literature? Especially, Is the finding that structural social capital is effective in choosing medical institutions while cognitive social capital is important in whether to seek treatment validated by the extant literature?
Response 1: Thank you for your questions regarding the theoretical contributions of this article, which also led the authors to revisit the theoretical innovations of the article.
The article has been revised in the literature review section for previous research on structural and cognitive social capital. The more authoritative and academically recognized research on the classification and measurement of structural and cognitive social capital is currently by Uphoff, so the article uses Uphoff's classification of social capital as the standard and is annotated in the text. See these paragraphs in the text for details.Meanwhile, the article concludes the study by stating that cognitive social capital mainly affects whether to seek medical treatment, and structural social capital mainly affects choice of medical institution. In the stage of deciding whether to seek medical treatment when rural residents are sick, the main influencing factor is cognitive social capital; while in the stage of choosing medical institutions, structural social capital plays a major role.
Point 2:How about the estimated coefficients of control variables (i.e., propensity factors)? How do they relate to the dependent variables? How do you evaluate the estimation results with regard to them? [522-523] & [433-434]
Response 2: Thank you for your reference to the issue of control variables, I acknowledge that an explanation of the relationship between the control variables and the dependent variable should have been included in the article, and then this was merely an oversight rather than a failure to provide a description of it for the sake of brevity of the article. This article has been revised and is specifically reflected in the article.
Point 3:The authors’ choice of medical institution includes: “1-Clinic, 2-Village Health Office, 3-Township Health Center, 4-Specialty Hospital, 5-General Hospital”. However, estimation results as to the multinomial logit model in Table 4 as to the influence of social capital on the choice of medical institution do not report results with regard to the clinic. Are they missing? or is there any other reason for their non-exclusion? [371-372] & [522-523]
Response 3: Thank you for pointing out the item "clinic" in the "choice of community institution". In this paper, regression analysis was conducted using the mlogit model, which can be considered as an estimation of multiple binary logit models with two pairs of different types of health care institutions in the explanatory variables. The explanatory variable ("community institution selection") is set as with values from 1 to 5, representing clinics, village health offices, township health centers, specialty hospitals, and general hospitals, in that order. Further, the probability of choosing other types of health care institutions compared to clinics was analyzed by selecting clinic (as the "reference group" with coefficient=0). Therefore, "clinics" as the "reference group" did not appear in the estimation results of the logit model.
Point 4:There are recurring questions designed to determine whether a given sample has religious beliefs: “Do you believe in God?” and it shall be removed accordingly. [320-326]
Response 4: Thank you very much for pointing out the error in my article writing. Because of my carelessness, I wrote the question "Do you believe in God?" twice. I apologize for writing this question twice, and I have removed one redundant statement in this revision.
Point 5:One of the criteria used to choose the sample is “had health problems in the past two weeks” which constitutes “patient”. What is the rationale behind the choice of the past two weeks? [282-291]
Response 5: Thank you very much for raising the concern about the "composition of the sample criteria". On the one hand, in the specific health problems surveyed, the health problems are connoted with symptoms of disease. On the other hand, the survey process is based on information that can be remembered by the respondent to ensure that the information is true and accurate, and may be forgotten if too much time has passed. The data are more convincing for the recent physical conditions and medical visits within two weeks of our study. In addition, we are using the questionnaire and data from CFPS2018, which is fully titled China Household Dynamics Tracking Survey. This project is a major social science project implemented by the China Social Science Survey Center of Peking University with the goal of providing data for academic research and policy decisions by tracking and collecting data at three levels: individual, household, and community, and responding to social, economic, demographic, educational, and health changes in China. The project has been implemented for 12 consecutive years with this fixed question, so its questionnaire and survey data have high reliability and stability.
Point 6: To my understanding, among the propensity factors- control variables- questions: “Did you smoke in the past month” and “Did you drink alcohol more than three times a week in the past month” are there to check whether a given sample has smoking and alcohol consumption habits? Would not it be better had the questions directly asked a given sample whether they smoke or consume alcohol on a regular basis? Smoking in the past month or drinking alcohol more than three times a week in the past month may not alone be able to lead to the need for medical service utilization on the part of a patient. [346-347]
Response 6: Thank you very much for your question about control variables. As a rule, minor disease symptoms last for a short period of time, but habit formation is something that takes a long time to last. Alcohol consumption and smoking are long-standing habitual matters. However, the overall study of this paper was designed to study the respondents' recent conditions, so the questions "Did you smoke in the past month" and "Did you drink alcohol more than three times a week in the past month" were used. week in the past month". In addition, this paper uses data from the CFPS fixed questionnaire, which is implemented by the China Social Science Survey Center of Peking University. The project uses computer-assisted survey technology to conduct interviews, and this questionnaire has been adopted for the past ten years to meet diverse design needs, improve interview efficiency, and ensure data quality.
Point 7:How do authors evaluate the relatively low values of Pseudo R² in Table 2 and Table 4? [433-434] & [522-523]
Response 7: Thank you for your concern and question about the goodness-of-fit values in the paper, which should also be clarified by the authors. Because regressions are divided into explanatory and predictive regressions, predictive regressions generally place more emphasis on goodness-of-fit values. In this paper, the focus is on the explanatory regression, which focuses on explaining the use of social capital for health care services, so the regression analysis in this paper focuses more on the overall significance of the model and the statistical significance and economic significance of the independent variables. The goodness of fit is only a measure of how well the sample regression function fits the sample data, and does not fully reflect the overall problem; it still depends on the significance test results of the variables.As can be seen from the paper, the variables to be tested in this paper ("cognitive social capital" and "structural social capital") basically fulfill the previous hypotheses of this paper.
In addition, since the focus of our study is on the impact coefficients and the goodness-of-fit is not a priority, we have removed it for the sake of focus and hereby explain it to you.
Point 8:The authors stated that the mean values as to variables included in the study are not statistically different [378-379]. Did the authors check whether statistically significant differences exist between included variables with respect to the two samples via any statistical test?
Response 8: Thank you for your sincere suggestion regarding the description of the data in the article. The authors originally intended to convey that the sample data did not differ significantly in their means across the two models, but really did not show its specific significance throughout the statistical study, so this statement was removed from the article in the revision.
We have carefully studied your comments and made corrections accordingly. Please find attached the original manuscript revised according to the reviewer's comments, and the revised part is marked with the "track changes" function for your quick review, please see the attachment for the detailed revision: Revised version ijerph-2012359
Thank you again for your valuable comments on my article, and I hope to receive your approval.
Yours sincerely,
Corresponding author:Jiqing Hong
E-mail: hej96@stu.zuel.edu.cn

Round 2
Reviewer 1 Report
Thanks for your feedback. Although the revised manuscript has been greatly improved, there are still some problems that must be further addressed.
1.The article only uses the 2018 CFPS, which is not enough to support the research conclusion. I must remind you that the full name of CFPS is China Family Panel Studies but is not the Chinese Household Dynamics Tracking Survey. What's important, In 2010, CFPS officially carried out a baseline survey in 25 provinces/municipalities/autonomous regions across the country, and the follow-up suvey is conducted every two years.
2.You analyze structural social capital and cognitive social capital, and emphasize that the purpose of the article is not to compare the difference between the two social capital, I want to know how cognitive social capital and structured social capital affect the utilization of rural residents' medical services, and which one is more important? Can you provide a basic theoretical analysis framework and mechanism?
Author Response
Dear reviewer,
Thank you very much for your approval of the revision and another constructive review of my article (ijerph-2012359). I gave serious thought of your queries and found that have not expressed my views clearly in the last response. I am sorry. The following is my detailed response to your comments.
Point 1: The article only uses the 2018 CFPS, which is not enough to support the research conclusion. I must remind you that the full name of CFPS is China Family Panel Studies but is not the Chinese Household Dynamics Tracking Survey. What's important, In 2010, CFPS officially carried out a baseline survey in 25 provinces/municipalities/autonomous regions across the country, and the follow-up survey is conducted every two years.
Response 1:
You are quite right, the full name of CFPS is China Family Panel Studies, which was indeed began in 2010, and the follow-up survey is conducted every two years.
This paper studies the impact of social capital on rural residents' medical service utilization. Because social capital is relatively stable and does not change significantly, it is not suitable to use data of multi years to study the trend or compare the difference impact of different years.
Therefore, we use cross-sectional data for a particular year. And because of the epidemic, the data before 2019 reflected the medical service utilization more accurately, the data of 2018 is therefore used. It is aimed to analyze different samples in the same time, exploring the impact of different social capital of rural residents, leading to differences of medical service utilization.
CFPS is authoritative and has high reliability, the results in table 2 and 3 show that the conclusion is effective.
Point 2: You analyze structural social capital and cognitive social capital, and emphasize that the purpose of the article is not to compare the difference between the two social capital, I want to know how cognitive social capital and structured social capital affect the utilization of rural residents' medical services, and which one is more important? Can you provide a basic theoretical analysis framework and mechanism?
Response 2:
We apologize for any misunderstanding caused by my unclear expression in the last response. This article studies the relationship between social capital and rural residents' medical service utilization, and it analyzes and compares how the two types of social capital affect the utilization of rural residents' medical service.
The result showed that cognitive social capital mainly influences whether to seek medical treatment and structural social capital mainly influences the choice of medical institutions. In the conclusion of the study, it is stated that the effects of cognitive social capital and structural social capital differ at different stages of medical service utilization. In the stage of deciding whether to seek medical treatment when rural residents are sick, the main influencing factor is cognitive social capital; while in the stage of choosing medical institutions, structural social capital plays a major role.
The above is my response to your comments, and I would like to express my sincere apologies to you again for misunderstanding. Thank you again for your valuable comments on my article, and I look forward to receiving your approval.
Best wish to you.
Yours sincerely,
Corresponding author:Jiqing Hong
- mail: hej96@stu.zuel.edu.cn
